# Identification and Quantitation of Novel *ABI3* Isoforms Relative to Alzheimer’s Disease Genetics and Neuropathology

**DOI:** 10.3390/genes13091607

**Published:** 2022-09-08

**Authors:** Andrew K. Turner, Benjamin C. Shaw, James F. Simpson, Steven Estus

**Affiliations:** Department of Physiology and Sanders-Brown Center on Aging, University of Kentucky, 789 S Limestone St., Lexington, KY 405361, USA

**Keywords:** Alzheimer’s disease, RNA splicing, genetics, polymorphism, microglia

## Abstract

Elucidating the actions of genetic polymorphisms associated with the risk of Alzheimer’s disease (AD) may provide novel insights into underlying mechanisms. Two polymorphisms have implicated *ABI3* as a modulator of AD risk. Here, we sought to identify *ABI3* isoforms expressed in human AD and non-AD brain, quantify the more abundant isoforms as a function of AD genetics and neuropathology, and provide an initial in vitro characterization of the proteins produced by these novel isoforms. We report that *ABI3* expression is increased with AD neuropathology but not associated with AD genetics. Single-cell RNAseq of APP/PS1 mice showed that *Abi3* is primarily expressed by microglia, including disease-associated microglia. In human brain, several novel *ABI3* isoforms were identified, including isoforms with partial or complete loss of exon 6. Expression of these isoforms correlated tightly with total *ABI3* expression but were not influenced by AD genetics. Lastly, we performed an initial characterization of these isoforms in transfected cells and found that, while full-length *ABI3* was expressed in a dispersed punctate fashion within the cytosol, isoforms lacking most or all of exon six tended to form extensive protein aggregates. In summary, *ABI3* expression is restricted to microglia, is increased with Alzheimer’s neuropathology, and includes several isoforms that display a variable tendency to aggregate when expressed in vitro.

## 1. Introduction

The heritability of Alzheimer’s disease (AD) is estimated to be 60 to 80% [1]. To identify the genetic variants that mediate this heritability, a series of genome-wide association studies (GWAS) have been performed. Among the single-nucleotide polymorphisms (SNPs) associated with AD risk, two SNPs have implicated the Abelson-interactor family member 3 (*ABI3*) gene, including a rare missense SNP (rs616338) within *ABI3* exon 5 and a common SNP (rs28394864) located 150,000 bp downstream of *ABI3* [2,3,4,5]. For each SNP, the minor allele is associated with increased AD risk.

*ABI3* is expressed primarily by microglia in the human brain [6,7]. Our understanding of the actions of microglia in AD is evolving, but microglia are currently believed to play a biphasic role in AD. Early in the AD process, microglia appear helpful because they contribute to the clearance of amyloid-beta which accumulates in AD [8]. However, late in the disease process, microglia appear deleterious because microglial activation and resulting inflammation appear to promote neurofibrillary tangle formation and neuronal death, which correlate with dementia [9,10,11]. 

ABI3 is a component of the WAVE (WASP-family verprolin homologous protein) complex that regulates actin polymerization [12]. The WAVE complex facilitates polymerization by recruiting cytosolic G-actin and integrating it into the cytoskeletal structure [13]. The three ABI family members, ABI1, ABI2 and ABI3, compete for integration into the WAVE complex [14]. While ABI1 and ABI2 both facilitate actin polymerization, ABI3 inhibits the WAVE complex and its ability to promote actin polymerization [14]. 

The prototypic *ABI3* gene has 8 exons, with exons 2 to 4 encoding the ABI-homologous region and exons 7 and 8 encoding an SH3 domain (Figure 1). The prototypic ABI3 protein consists of 366 amino acids, and may be regulated by phosphorylation, noting that S213, S216, Y341 and S342 can be phosphorylated [15,16,17]. Additionally, the missense SNP that is associated with AD, rs616338 (exonic C>T), changes S209 to a phenylalanine. This SNP increases the risk of AD (odds ratio = 1.43) and is rare, with a 1% minor allele frequency (MAF) in Europeans and less abundant in other populations [2]. The second AD-associated SNP, rs28394864 (genomic G>A), also increases AD risk (odds ratio = 1.012) but is more common, with a MAF ranging from 22 to 47% in Africans to Europeans, respectively [4]. Although these SNPs are associated with AD risk, studies of AD-related phenotypes have not yet identified pathologies such as amyloid or tau levels that are consistently and robustly associated with either of these SNPs [18,19,20]. Interestingly, rs28394864 has been associated with cardiovascular traits including hypertension, ischemic heart disease, and coronary atherosclerosis [21].

Here, to test the hypothesis that AD genetics or neuropathology affects *ABI3* expression and splicing, we identified and quantified *ABI3* isoforms in a series of well-characterized AD and non-AD brains and genotyped the samples for rs616338 (S209F) and rs28394864. We found that *ABI3* expression is increased with AD neuropathology but not genetics. By using single-cell RNAseq to evaluate a mouse model of amyloid accumulation, we confirmed that *Abi3* is expressed in microglia, including disease-associated microglia. We identified several novel *ABI3* isoforms, with exon 6 having the most variation. Lastly, isoforms lacking exon 6 encoded proteins that showed a variable tendency to aggregate when expressed in HMC3 microglial cells.

## 2. Materials and Methods

### 2.1. Human Brain Samples

Genomic DNA and cDNA samples used for this study have been extensively described previously [22,23,24,25]. Briefly, RNA and genomic DNA were purified from anterior cingulate samples provided by the University of Kentucky AD Center, and the RNA was converted to cDNA as described [22,23,25]. Samples were divided into two groups based on AD neuropathology, as quantified by the National Institute on Aging–Reagan Institute (NIARI) criteria, which include both neuritic plaques and neurofibrillary tangles [26]. Samples from brains with a NIARI score of 3 have frequent plaques and tangles in the neocortex and were considered high pathology (n = 26). Samples with a NIARI score of 0–2 have low-to-moderate levels of plaques and tangles and were considered low pathology (n = 27). The age at death of the high pathology individuals was 82.0 ± 6.4 (mean ± SD), and that of the low pathology individuals was 82.4 ± 8.8. The postmortem intervals for the high and low pathology samples were similar, 3.5 ± 0.6 and 2.7 ± 0.9 h, respectively. Individual metadata are included within Appendix A.

### 2.2. PCR

Genomic DNA samples were genotyped for rs616338 and rs28394864 using TaqMan Genotyping (ThermoFisher, Waltham, MA, USA), as directed by the manufacturer: initial genomic denaturation at 95 °C for 10 min, and PCR cycling at 95 °C, 15 s; 60 °C, 1 min; 40 cycles. To identify *ABI3* splice variants in human brain, cDNA samples corresponding to 30 ng of RNA from six individuals were subjected to polymerase chain reaction (PCR) with primers corresponding to sequences within exon 1 and exon 8, 5′-ACAGAGGCTGGGGGTGAT and 5′-GGGAGGAAGGACAGAACACA, respectively. After 32 cycles with Platinum Taq DNA Polymerase (Thermo), PCR products were separated by electrophoresis on a 10% polyacrylamide gel and visualized with SYBR Gold staining. The PCR product, which contained multiple bands from 900 to 1100 bp, was excised in bulk, eluted from the acrylamide gel, reamplified in aggregate and then cloned into pcDNA3.1 by using TOPO-TA cloning (Thermo). A total of 64 random clones were sequenced (ACGT Inc., Chicago, IL, USA). To visualize isoforms in the vicinity of exon 6, primers corresponding to sequences within exon 5 (5′-GCCTCCTCTGCGTTTTCC) and exon 7 (5′-CTCATCAGGCCCAAATCCT) were PCR-amplified, separated on a 10% acrylamide gel and detected by using SYBR Gold fluorescence. 

### 2.3. qPCR

To quantify *ABI3* and its isoforms, a series of qPCR assays using PerfeCTa SYBR Green master mix were performed, as previously described [24]. For each assay, copy numbers present in cDNA samples were determined relative to standard curves executed in parallel [27]. Total *ABI3* was quantified by using primers corresponding to sequence within exons that were constitutively present, i.e., exons 1 and 2: 5′-TGAGGGGCAACCACAGT-3′ and 5′-CGCCTTCCGCTTGTCTG-3′, respectively, and PerfeCTa SYBR Green master mix, as previously described [24]. Isoforms retaining the first 69 bp of exon 6 were quantified by using primers corresponding to sequences within the first 69 bp of exon 6 (5′-GCAGGCAGCACCTCCAG) and within exon 7 (5′-GCCCCAATTCATCTCCATCCA). The retention of *ABI3* intron 6 was quantified with primers corresponding to sequences at the exon 4–5 junction (5′-GCCACCTTGGGGAGACCAC) and intron 6 (5′-CACGGTTGGGAAGAAGAGGGT). Quantitation of *D-3bp ABI3*, which lacks the first three bp of exon 6, required the use of a TaqMan assay that used primers corresponding to sequences within exon 5 (5′-GACGGCAGACTCTCCGC) and exon 6 (5′-GGAGAGGTGGGGCTGGA) and a locked nucleic probe corresponding to the novel exon 5–exon 6 junction (56-FAM/GGCCGG+C+A+GCGCCGA/3IABkFQ, IDT). Cycling conditions for all qPCR were as follows: 95 °C, 2 min; 95 °C, 15 s, 60 °C, 15 s, 72 °C, 30 s, 40 cycles. The expression of *ITGAM* and *AIF-1* was quantified, as described previously [27].

### 2.4. Mice

APP/PS1 (APPswe, PSEN1dE9) mice express a chimeric mouse/human amyloid precursor protein (Mo/HuAPP695swe) and a human *PSEN1* gene lacking exon 9 (PS1-dE9) [28]. These mice begin to develop Aβ deposits by six months of age, with abundant plaques in the hippocampus and cortex by nine months [28]. Mice were bred by crossing *APP/PS1* carriers with wild-type C57Bl/6J mice. Non-APP/PS1 (WT) littermates served as control mice. Mice were maintained on standard mouse chow (Teklad Global 18% Protein Rodent Diet) in individually ventilated cages. Beginning at five months of age, drinking water for both WT and APP/PS1 mice was supplemented with sodium chloride (132.5 mM) [29]. This solution was made fresh weekly and administered via water bottles until the mice were euthanized at 10 months of age. 

### 2.5. Single-Cell RNA Seq

Brain tissues were processed into single-cell suspensions, as previously described [30]. Briefly, 10-month-old female APP/PS1 or WT mice (pooled n = two per genotype) were anesthetized with 5.0% isoflurane, perfused with ice-cold Dulbecco’s phosphate-buffered saline, and right cerebellar hemispheres quickly isolated and minced. Tissue was dissociated by using Adult Brain Dissociation Kit (ADBK) enzymatic digest reagents (Miltenyi, Gaithersburg, MD, USA) and the “37C_ABDK” protocol on the gentleMACS Octo Dissociator instrument (Miltenyi) with heaters attached. The resulting cell suspensions were filtered sequentially through one 70 μm and two 30 µm mesh cell filters. Cell viability (AO/PI viability kit, Logos Biosystems) was >88%. Diluted cell suspensions (10 cell/uL) were loaded onto the 10× Genomics Chromium Controller and libraries prepared using the Chromium v3 Single Cell 3′ Library and Gel Bead Kit (10× Genomics). Final library quantification and quality check were performed using BioAnalyzer (Agilent), and sequencing was performed on NovaSeq 6000 S4 flow cell, 150 bp Paired-End sequencing (Novogene, Sacramento, CA, USA). Raw sequencing data were de-multiplexed and aligned using Cell Ranger (10× Genomics), and further processed using Partek software (Build 10.0.22.0315, Chesterfield, MO, USA) as described previously [31]. To remove likely multiple and dead cells, cells were discarded if they had total unique molecular identifiers of less than 5000 or greater than 50,000, if total features per barcode were less than 1800 or more than 5000, or if mitochondrial read counts were more than 30% of total read counts. Apparent microglia were assigned by using *Tmem119* expression, and disease-associated microglia were assigned by using *Lpl* expression [27]. 

### 2.6. Expression Cloning of ABI3 Isoforms

To evaluate the expression of the *ABI3* isoforms in transfected cells, each isoform was cloned in frame after the carboxyl terminus of GFP by using NT-GFP Fusion-TOPO, as directed by the manufacturer (Thermo). The identity of the fusion proteins was confirmed by sequencing (ACGT, Inc.). Human microglial clone 3 (HMC3) cells were transfected with these plasmids by using Lipofectamine^TM^ 3000 Reagent (Invitrogen, Waltham, MA, USA), as directed by the manufacturer. Two days later, cells were fixed with formalin, labeled with Alexa Fluor 568 Phalloidin by using the manufacturer’s protocol (ThermoFisher, Waltham, MA, USA), and mounted with Prolong Glass with NucBlue (ThermoFisher). Cells were then visualized by using confocal microscopy (Nikon A1R HD, Melville, NY, USA) and representative images were obtained.

### 2.7. Statistics

To generate a normal distribution of gene expression copy numbers, as determined by qPCR, we used the square root of the copy numbers. One sample with a *D-Exon6 ABI3* that was greater than the mean value plus six times the standard deviation of all samples was not used in these calculations. The effect of AD neuropathology and genetics on total *ABI3* expression was then evaluated by comparison with the geometric mean of *AIF-1* and *ITGAM* expression by using a linear regression model (SPSS vs. 28). The effect of AD neuropathology and genetics on *ABI3* isoforms was evaluated by comparing each isoform’s expression relative to total *ABI3* expression also by using a linear regression model (SPSS vs. 28). 

## 3. Results

### 3.1. Identification of ABI3 Isoforms in Human Brain

To identify *ABI3* isoforms present in human brain, we performed PCR with primers corresponding to sequences in exon 1 and exon 8 (Figure 1). The resulting PCR products were cloned and 64 random clones were sequenced, resulting in the identification of multiple novel isoforms (Figure 2A). The three most common isoforms involved exon 6 and included (i) an isoform wherein the first three base pairs (bp) of exon 6 are retained with intron 5 (referred to as *D-3bp*), resulting in the loss of Ser216; (ii) an isoform lacking the first 69bp of exon 6 (referred to as *D-69bp*), which removes the first 23 amino acids encoded by exon 6; and (iii) an isoform lacking exon 6 entirely (referred to as *D-Exon 6*). Since prototypic exon 6 is 158bp, exclusion of exon 6 results in a codon reading frameshift. Hence, D-Exon 6 lacks the 52 amino acids encoded by exon 6, with the frameshifted exon 7 encoding 18 novel amino acids followed by a premature stop codon. Additional isoforms were present and appeared to be less abundant (Figure 2A). Since the exon 6 variants were the most abundant and the primary AD-associated *ABI3* SNP, rs616338 (S209F), resides in exon 5, only 19 bp from exon 6, we focused on the exon 6 isoforms. Hence, we performed PCR with primers corresponding to exon 5 and exon 7 on a series of AD and non-AD brain samples (Figure 2B). These results suggested that each isoform was present regardless of disease status. 

### 3.2. Quantitation of ABI3 Isoform Expression as a Function of AD Neuropathology and Genetics

To quantify *ABI3* expression as a function of AD neuropathology and genetics (rs28394864 and rs616338), qPCR was performed on cDNA prepared from 53 brain samples. Total *ABI3* expression was determined by performing qPCR with primers corresponding to sequences within the constitutively expressed exon 1 and exon 2. Since Satoh et al. have reported that ABI3 protein expression is restricted to microglia [6], we compared *ABI3* expression with the geometric mean of the microglial genes *AIF-1* and *ITGAM. ABI3* expression strongly correlated with microglial gene expression (*p* < 0.0001, r^2^ = 0.784) and was modestly associated with AD neuropathology (*p* = 0.02, F_1,51_ = 5.77), but not with either AD-associated SNP (*p* > 0.05, Figure 3A–C).

We proceeded to quantify *ABI3* isoforms that appeared the most common based on the clone counts. Expression of the *D-3bp* isoform correlated with total *ABI3* expression (*p* < 0.0001, r^2^ = 0.584, Figure 4A). Relative to total *ABI3* expression, *D-3bp* expression was independent of both AD neuropathology (*p* > 0.05, Figure 4A) and genetics (*p* > 0.05, Figure 4B,C). We next performed qPCR to quantify *ABI3* expression containing the first 69 base pairs of exon 6 (referred to as normal exon 6). *ABI3* normal exon 6 expression strongly correlated with total *ABI3* expression (*p* < 0.0001, r^2^ = 0.916, Figure 4D). Expression of this isoform was also independent of both AD neuropathology (*p* > 0.05, Figure 4D) and genetics (*p* > 0.05, Figure 4E,F). The isoform lacking exon 6 entirely, *D-Exon6,* had more variation in its expression relative to total *ABI3* (*p* = 0.051, r^2^ = 0.167) than the other isoforms (Figure 4G). *D-exon6* expression was not significantly influenced by AD neuropathology or genetics (*p* > 0.05, Figure 4G–I). During the course of this work, we identified another novel *ABI3* isoform that retained intron 6. This isoform encodes prototypic ABI3 through exon 6 but has a premature stop codon located within intron 6. This isoform was likely derived from RNA and not contaminating genomic DNA, because intron 5 was not present. When we quantified the expression of this isoform, we found that it was rare, correlated well with total *ABI3* expression (*p* < 0.0001, r^2^ = 0.702, Figure 4G), but was also not influenced by either AD neuropathology (*p* > 0.05, Figure 4G) or AD genetics (*p* > 0.05, Figure 4H,I).

### 3.3. ABI3 Expression in Homeostatic and Disease-Associated Microglia

Since *ABI3* expression was increased with AD neuropathology, we hypothesized that *ABI3* may be expressed in disease-associated microglia that accumulate in AD and AD models [32]. To test this hypothesis, we performed a single-cell RNAseq on APP/PS1 mice that were 10 months of age, when disease-associated neuropathology is abundant [33]. The population of microglia, detected as expressing *Tmem119*, included a cluster present in the APP/PS1 but not wild-type mice (Figure 5A). Analysis of *Lpl* expression, a marker of disease-associated microglia [32], confirmed that this cluster was disease-associated microglia (Figure 5B). *Abi3* expression was robust in all microglia, and especially in disease-associated microglia (Figure 5C). 

### 3.4. ABI3 Isoform Expression In Vitro

To better understand the potential impact of the variant ABI3 isoforms, the cellular location of the more abundant isoforms was visualized by expressing them as a fusion protein with GFP at their amino-terminus, as has been described previously for ABI3 [34]. Although ABI3 regulates actin polymerization, it does not colocalize with actin, instead appearing in a disperse cytosolic localization. When these isoforms were transfected into HMC3 microglial cells, the expression pattern of the full-length (Figure 6A), D-3bp (Figure 6B), and D-69 (Figure 6C) ABI3 isoforms appeared very similar, manifesting as fine puncta with a dispersed cytosolic localization. A dispersed cytosolic localization for ABI3 fusion proteins has been described previously [14], and GFP fusion proteins often appear as similar puncta [35,36]. In contrast to the patterns observed with the other isoforms, the D-Exon6 (Figure 6D) isoform primarily manifested as unique “ribbon” formations or very large aggregates.

## 4. Discussion

This study had several primary findings. First, *ABI3* expression is increased in the brains of individuals with high AD neuropathology. Some of this increase may be related to ABI3 expression in disease-associated microglia, which are increased in AD. Second, *ABI3* expression was not associated with two SNPs associated with AD risk, rs28394864 or rs616338. Third, multiple novel *ABI3* splice variants were identified, including several involving exon 6. Their expression correlated tightly with total *ABI3* expression but not with AD genetics. Lastly, when these novel isoforms were expressed in transfected cells, FL-ABI3, D-3bp, and D-69bp were expressed in a dispersed punctate fashion within the cytosol, while the D-Exon6 isoform tended to localize into larger aggregates and unique ribbon formations. In summary, *ABI3* expression is increased with AD neuropathology but not AD genetics. Further studies are required to elucidate the role of the novel *ABI3* isoforms identified here.

AD genetic risk factors have been an area of intense scrutiny, because they may reveal mechanisms that represent drug targets to reduce AD risk. Two SNPs have focused attention on *ABI3*, including the missense SNP rs616338 and as well as rs28394864, which is in the vicinity of *ABI3* [2,3,4,5]. Here, we investigated the hypothesis that *ABI3* expression or splicing is influenced by rs28394864 and/or rs616338. Our findings did not support our hypothesis, although we did observe a modest trend that the rs616338 minor allele was associated with a decrease in *D-3bp* and intron-6 retention (*p* = 0.12 and 0.13, respectively). Note, however, that our statistical power to discern this association was limited because of the low number of rs616338 minor allele carriers in our dataset. At this time, the actions of both SNPs are unclear. However, the substitution of a Phe for a Ser at 209 was recently found to affect ABI3 phosphorylation [7]. Regarding rs28394864, *ABI3* was apparently identified as the best candidate gene for this SNP, because *ABI3* had previously been implicated in AD by S209F and is in the vicinity of rs28394864 [2,4,5]. It is possible that rs28394864 may act upon genes other than *ABI3* in this locus, noting that rs28394864 is associated with expression of both *GNGT2* and *ZNF652*, at least in some tissues, and that both of these genes are expressed in the brain [37,38,39].

The function of the novel ABI3 isoforms identified here is not clear. The main change in D-3bp ABI3 is the loss of Ser216, which is reported to undergo phosphorylation [17]. The D-69bp isoform also lacks Ser216 as well as 22 additional amino acids. However, since D-3bp and D-69bp displayed a localization pattern in transfected cells that was similar to FL-ABI3, and the 23 amino acids lost by D-69bp are poorly conserved (Figure 7), we speculate that both isoforms may retain function. In contrast, the D-Exon6 isoform shows a pattern of very robust protein aggregation, and, uniquely to this isoform, apparent protein ribbons. While GFP-tagged proteins have previously been reported to form aggregates [35,36], we posit that it is unlikely that these ribbon aggregates were solely due to overexpression or the GFP-tag, because the other isoforms did not demonstrate these ribbon formations. Other intracellular protein aggregates, such as neurofibrillary tangles and TDP43, occur in AD. However, D-Exon6 expression did not significantly differ by AD neuropathology or by AD genetics. Thus, it is unlikely that D-Exon6 affects AD. We do speculate that the D-Exon6 isoform represents a loss of ABI3 function, given that the D-Exon6 isoform results in a frame shift and early stop codon during translation and the protein’s unique tendency to aggregate, as compared to the other isoforms.

Considering the overall role of ABI3 in AD, we note that (i) the S209F SNP is associated with increased AD risk [2,3,18] and reduced ABI3 phosphorylation [7], and (ii) reduced ABI3 phosphorylation appears to increase ABI3 function [17]. Hence, we hypothesize that AD risk may be promoted by increased ABI3 function. This model could be tested in a murine model comparing both alleles of S209F in *ABI3*. Although results from this approach are not yet available, two studies with murine *Abi3* knockout models have been reported [7,40]. Both involve beta amyloid accumulation in a model that includes deletion of the *Abi3* locus, which also contains *Gngt2*. Unfortunately, these studies yielded inconsistent results, with Karahan et al. finding that *Abi3* locus deletion resulted in increased amyloid burden in 5xFAD mice, while Ibanez et al. found that *Abi3* locus deletion reduced amyloid burden in TgCRND8 mice [7,40]. Interestingly, Ibanez et al. also found that *Abi3* locus deletion exacerbated tau accumulation and inflammation in a murine model of tau aggregation [7]. Given these conflicting results, evaluation of amyloid and tau changes in a murine model involving S209F may be necessary to clarify the role of ABI3 in AD.

## 5. Conclusions

This study represents an initial discovery and characterization of novel *ABI3* isoforms. This study also found an increase in total *ABI3* expression with AD neuropathology. Further research is necessary to determine how the increase in *ABI3* expression in AD impacts AD progression and whether the novel *ABI3* isoforms contribute to AD pathogenesis. This research should include characterization of their impact on microglial functionality. Overall, these studies add to our understanding of *ABI3* expression, splicing, and subcellular localization as a function of AD neuropathology, genetics and microglial status.

## Figures and Tables

**Figure 1 genes-13-01607-f001:**
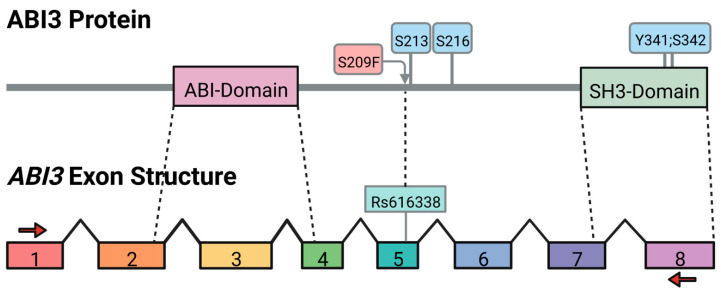
**ABI3 protein and gene structure**. The two primary domains of ABI3 are indicated along with phosphorylation sites (blue boxes). The site of rs616338 is indicated as S209F (red box). The exons are drawn to scale. The sites of PCR primers in exons 1 and 8 that were used to amplify *ABI3* isoforms are also indicated (red arrows).

**Figure 2 genes-13-01607-f002:**
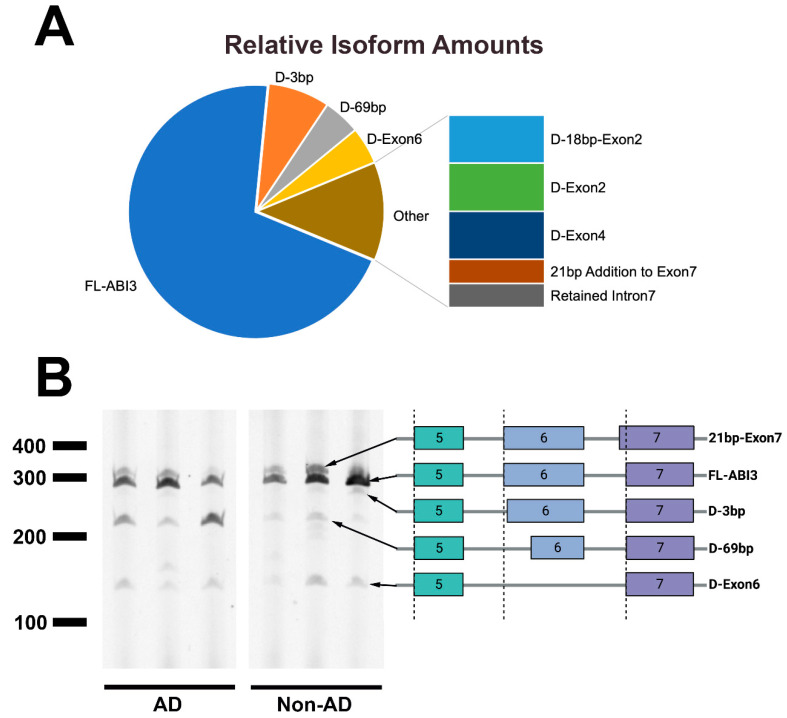
***ABI3* isoforms present in human brain.** Common *ABI3* isoforms were identified by PCR amplification from exons 1 to 8 and sequencing random clones. In addition to the exon 6 isoforms, other isoforms include a loss of 18bp at the end of exon 2 (*D-18bp-Exon2*), a deletion of exon 2 (*D-Exon2*), a deletion of exon 4 (*D-Exon4*), a 21 bp addition to the beginning of exon 7 (*21bp Addition to Exon 7*), and a retention of intron 7 (*Retained Intron 7*) (**A**). Multiple exon 6 isoforms are present in AD and non-AD brain cDNA. The structure of each of these isoforms from exons 5 through 7 is indicated (**B**).

**Figure 3 genes-13-01607-f003:**
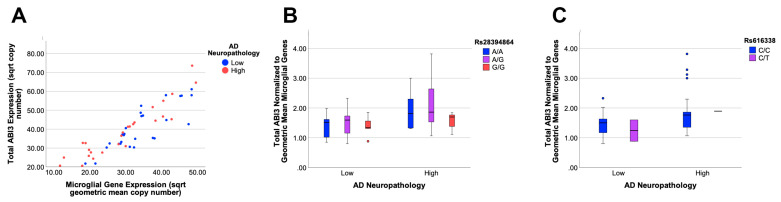
**Total *ABI3* expression increases with high AD neuropathology.***ABI3* expression correlates strongly with the expression of microglial genes, which are modeled here as the geometric mean of *AIF-1* and *ITGAM* expression (**A**). *ABI3* expression in samples with high AD neuropathology was higher than those with low neuropathology (*p* = 0.02). The number of samples with high and low pathology was 26 and 27, respectively. *ABI3* expression was not modulated by either SNP. Dots represent outliers which are 1.5 times more than the interquartile range. (**B**,**C**). The number of samples for each genotype were: rs2894864 (G/G = 16, G/A = 23, A/A = 14), and rs616338 (C/C = 50, C/T = 3).

**Figure 4 genes-13-01607-f004:**
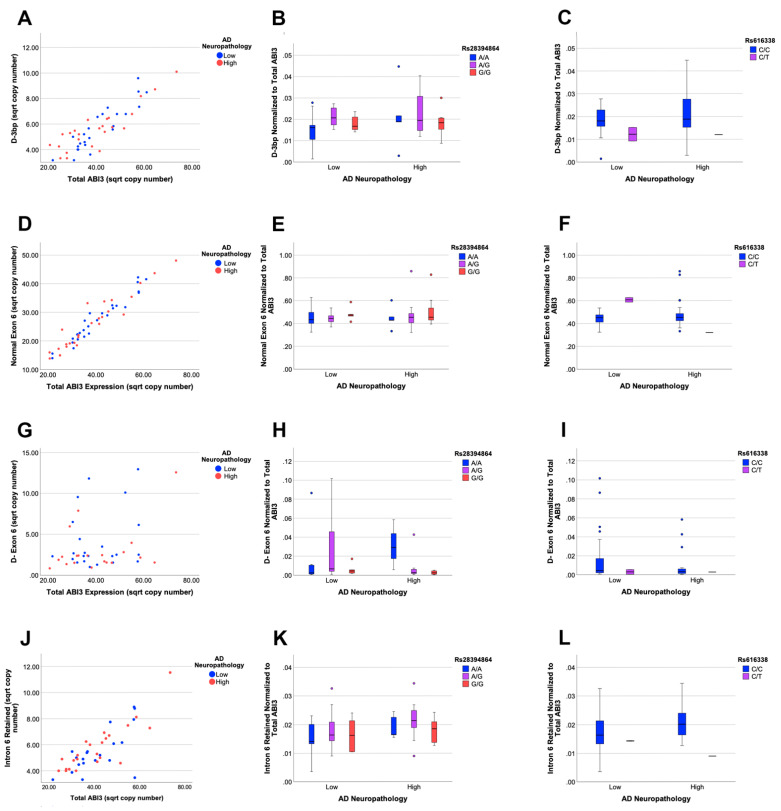
***ABI3* isoform expression is not influenced by AD neuropathology or genetics.** The expression of the *D-3bp* isoform correlates with expression of total *ABI3* (**A**). *D-3bp* expression was not significantly influenced by AD neuropathology or either SNP, but did trend with rs616338 (**B**,**C**). The expression of *ABI3* with prototypic exon 6 strongly correlates with the expression of total *ABI3* (**D**) but was not significantly influenced by AD neuropathology or either SNP (**E**,**F**). Expression of *ABI3* lacking exon 6 showed more variability than the other isoforms but also was not significantly influenced by AD neuropathology or genetics (**G**–**I**). Expression of *ABI3* with retained intron 6 correlated well with the expression of total *ABI3* (**J**). This isoform was not significantly influenced by AD neuropathology or either SNP, but did trend with rs616338 (**K**,**L**).

**Figure 5 genes-13-01607-f005:**
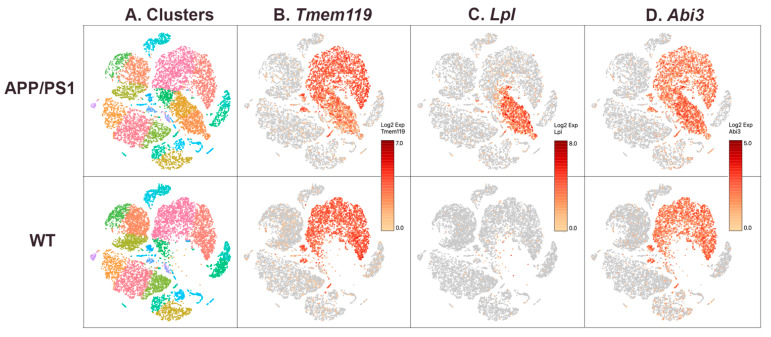
***ABI3* expression is largely restricted to microglia, including apparent disease-associated microglia.** Individual cells were clustered based on Loupe Browser analysis of transcript profiles and are color-coded to show these groups (**A**). Microglia were identified by virtue of *Tmem119* expression (**B**). Disease-associated microglia were identified by using *Lpl* expression (**C**). Note that this subpopulation of microglia is present in the APP/PS1 but not WT mice (**C**). *Abi3* was expressed in the same cell populations as *Tmem119*, including the disease-associated microglial population (**D**).

**Figure 6 genes-13-01607-f006:**
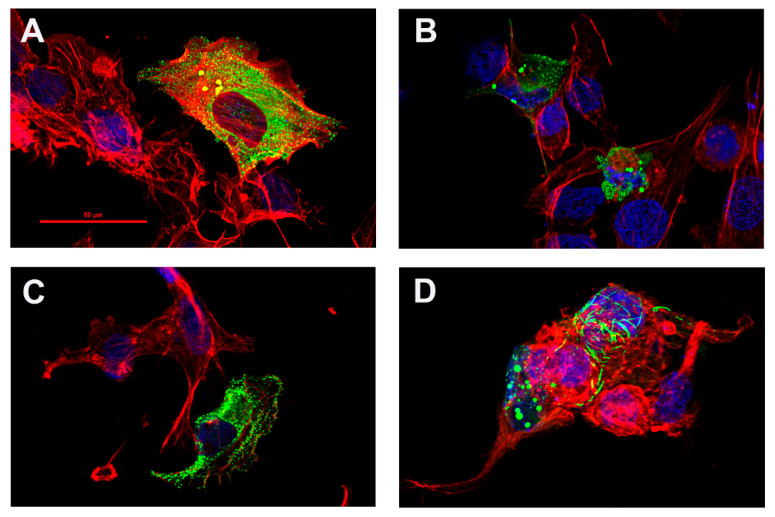
**ABI3-GFP expression in HMC3 cells results in “ribbon” formations with the D-Exon6 isoform.** Each isoform was expressed as ABI3-GFP fusion proteins (green) in HMC3 cells co-labelled with phalloidin (red) and DAPI (blue). The FL-ABI3 (**A**), D-3bp (**B**), and D-69bp (**C**) isoforms manifested in cells as small, dispersed puncta. The D-Exon6 (**D**) isoform formed large aggregates and unique “ribbon” formations often. Color-separated images for this figure are included as Appendix A.

**Figure 7 genes-13-01607-f007:**
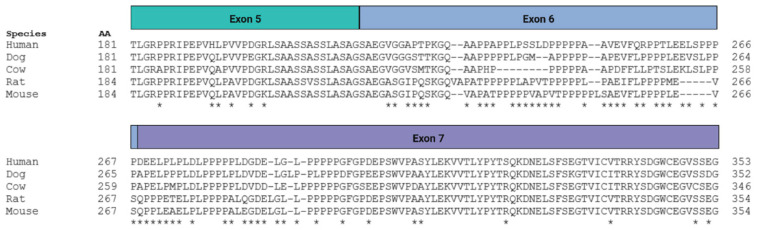
**Exon 6 is poorly conserved across mammalian species**. The prototypic ABI3 amino acid sequence from exons 5 to 7 was compared to its corresponding sequences in dog (*Canis lupus familiaris*), cow (*Bos taurus*), rat (*Rattus rattus*), and mouse (*Mus musculus*). Mismatches are indicated by asterisks. Note that gaps due to alignment did not count as mismatches unless the other amino acids present also varied. The amino acid sequence encoded by exon 6 was poorly conserved, with many of its amino acids diverging across other mammals.

## Data Availability

Raw data from the single-cell RNA sequencing study will be uploaded to NIH GEO, a public functional genomics data repository.

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
