# Peer review of "Identification and Quantitation of Novel ABI3 Isoforms Relative to Alzheimer’s Disease Genetics and Neuropathology"

_genes, 2022, doi:10.3390/genes13091607_

Round 1

Reviewer 1 Report

Turner et al. have applied several functional analyses to elucidate the mechanism of action of the previously identified SNPs in ABI3 that are associated with AD risk. This study adds up to our understanding of the mechanism of action of ABI3 isoforms in the context of AD and can provide novel insights into molecular mechanisms of AD pathogenesis. Overall, this is an informative study but could benefit from a few improvements:

1) Page 2, Introduction: The S209 mutant has been characterized before and indicated to affect phosphorylation of Abi3 (PMID: 35897046). The authors have cited this work in their discussion, but as they’re introducing different SNPs of ABI3, it would be useful to have a few sentences about the S209 reported phenotype as well (in the intro or discussion).

2) If known, could the authors indicate odds ratio for all SNPs including the rare allele (S209)?

3) Page 9, line 279: Please reword the first sentence, it is ambiguous and unclear.

4) It could be useful if the results section is divided into sub-sections for organizational purposes.

5) The authors indicate having normalized ABI3expression levels to AD neuropathology. However, it is not clear what kind of “AD neuropathology” they refer to. Neither is it clear how the scoring has been done. The authors should consider thoroughly defining what they mean by neuropathology. Are these pathology assessment on human brains or just clinical assessments of the patients or both? If so, what exactly have been assessed and considered for scoring in these analyses (Aβ, tau, neuronal loss, etc.?)

6) Regard the aggregates: The authors suggest some isoforms of ABI3 are prone to form aggregates based on microscopy analysis on GFTP-tagged proteins they have generated in their cell model. However, the authors should consider excluding two other likely scenarios that could lead to formation of aggregate/puncta: 1) They’re overexpressing ABI3 isoforms (they’re using CMV promoter as per their methods and this results in highly overexpressed levels of protein). It would be helpful if the levels were quantified relative to normal physiological levels. 2) GFP-tagged proteins have a tendency to form aggregates due to an artifact caused by the protein labeling (PMID: 32901052, PMID: 34687761 and PMID: 32677995 as examples). The authors should consider trying another smaller tag (e.g. Myc) to make sure the aggregate formation is a native behavior of their protein rather than an effect caused by GFP. 

7) The authors are not fully clear about the impact or conclusive points they have on aggregate formation by D-Exon6? Is this something they believe contributing to the function of this isoform in the context of AD?

8) Lines 334-336: reference 7 (Ibanez et al.) have looked at the impact of loss of Abi3-Gngt2, not just Abi3.This needs to be considered when comparing the two studies.

Minor edits:

There are some minor mistakes in the text, and some sentences that could be improved towards reading better. Examples:

Page 2, Line 40: “role as an ABI family member”

Page 6, Line 204: “but not with any of the AD associated genes”

Page 2, Line 43: “The WAVE complex acts by recruiting G-actin for polymerization and its integration into F-actin”.

Reviewer 2 Report

Thank you for the opportunity to review this manuscript, dealing with interesting findings entitled “Identification and quantitation of novel ABI3 isoforms relative to Alzheimer’s disease genetics and neuropathology”. All findings are interesting, and the article includes a balanced and critical view of the findings. However, their data representation is poor in addition to the lack of labeling in the scheme/Figure and low graphic image presentation. Figure 1 is good but is incomplete and does not sound good for the general readers. The author needs to redraw it with good quality graphics and proper labeling. The representation of Figure 5 is fine. The author must upload Figures 3 and Figure 4 with good graphics and larger font size. The author needs to add AD and age match control samples information in a table with PMI, Braak stages, sex, and Neurotic plaques information if available. It would be nice to present Figure 6 with both probe images and

colocalization. In addition, some citations are too long (such as citation 2), it would be nice if the author can replace these citations with others if possible. Furthermore, the author also needs to edit the sample size wherever necessary as some data does not show it. It is recommended to draw the overall study design in a scheme to make readers understand.

Round 2

Reviewer 2 Report

N/A